

# Green synthesis of silver nanoparticles using *Ocimum sanctum* Linn. and its antibacterial activity against multidrug resistant *Acinetobacter baumannii*

Deepan Gautam[1,2], Karma Gurmey Dolma[2], Bidita Khandelwal[3], Madhu Gupta[4], Meghna Singh[4], Tooba Mahboob[1], Anil Teotia[5], Prasad Thota[5], Jaydeep Bhattacharya[6], Ramesh Goyal[7], Sonia M.R. Oliveira[8,9], Maria de Lourdes Pereira[9,10], Christophe Wiart[11], Polrat Wilairatana[12], Komgrit Eawsakul[13], Mohammed Rahmatullah[14], Shanmuga Sundar Saravanabhavan[15] and Veeranoot Nissapatorn[1]

[1] School of Allied Health Sciences and World Union for Herbal Drug Discovery (WUHeDD), Walailak University, Nakhon Si Thammarat, Thailand

[2] Department of Microbiology, Sikkim Manipal Institute of Medical Sciences, Sikkim Manipal University, Gangtok, Sikkim, India

[3] Department of Medicine, Sikkim Manipal Institute of Medical Sciences, Sikkim Manipal University, Gangtok, Sikkim, India

[4] Department of Pharmaceutics, School of Pharmaceutical Sciences, Delhi Pharmaceutical Sciences and Research University, New Delhi, India

[5] Department of Microbiology, Indian Pharmacopoeia Commission, Ministry of Health and Family Welfare, Ghaziabad, Utter Pradesh, India

[6] School of Biotechnology, Jawaharlal Nehru University, New Delhi, India

[7] Department of Pharmacology, Delhi Pharmaceutical Sciences and Research University, New Delhi, India

[8] Hunter Medical Research Institute, New Lambton, Australia

[9] CICECO-Aveiro Institute of Materials, University of Aveiro, Aveiro, Portugal

[10] Department of Medical Science, University of Aveiro, Aveiro, Portugal

[11] The Institute for Tropical Biology and Conservation, University Malaysia Sabah, Sabah, Malaysia

[12] Department of Clinical Tropical Medicine, Faculty of Tropical Medicine, Mahidol University, Bangkok, Thailand

[13] School of Medicine, Walailak University, Nakhon Si Thammarat, Thailand

[14] Department of Biotechnology & Genetic Engineering, University of Development Alternative, Dhaka, Bangladesh

[15] Department of Biotechnology, Aarupadai Veedu Institute of Technology, Vinayak Mission's Research Foundation (DU), Chennai, Tamil Nadu, India

Corresponding authors
Karma Gurmey Dolma,
kgdolma@outlook.com
Veeranoot Nissapatorn,
nissapat@gmail.com

## ABSTRACT

The biosynthesis of nanoparticles using the green route is an effective strategy in nanotechnology that provides a cost-effective and environmentally friendly alternative to physical and chemical methods. This study aims to prepare an aqueous extract of *Ocimum sanctum* (*O. sanctum*)-based silver nanoparticles (AgNPs) through the green route and test their antibacterial activity. The biosynthesized silver nanoparticles were characterised by colour change, UV spectrometric analysis, FTIR, and particle shape and size morphology by SEM and TEM images. The nanoparticles are almost spherical to oval or rod-shaped with smooth surfaces and have a mean particle size in the range of 55 nm with a zeta potential of $-2.7$ mV. The antibacterial activities of AgNPs evaluated against clinically isolated multidrug-resistant *Acinetobacter baumannii* (*A. baumannii*) showed that the AgNPs from *O. sanctum* are effective in inhibiting *A. baumannii* growth with a zone of inhibition of 15 mm in the agar well diffusion method and MIC and

MBC of 32 μg/mL and 64 μg/mL, respectively. The SEM images of *A. baumannii* treated with AgNPs revealed damage and rupture in bacterial cells. The time-killing assay by spectrophotometry revealed the time- and dose-dependent killing action of AgNPs against *A. baumannii,* and the assay at various concentrations and time intervals indicated a statistically significant result in comparison with the positive control colistin at 2 μg/mL ($P < 0.05$). The cytotoxicity test using the MTT assay protocol showed that prepared nanoparticles of *O. sanctum* are less toxic against human cell A549. This study opens up a ray of hope to explore the further research in this area and to improve the antimicrobial activities against multidrug resistant bacteria.

# INTRODUCTION

The synthesis of nanoparticles through the green route is a growing subject in nanotechnology that offers cost-effective and environmentally friendly alternatives to traditional physical and chemical processes (*Yeo, Lee & Jeong, 2003*). Although nanoparticles are the most promising emerging formulation designs, among these silver nanoparticles are outstanding owing to their all-around pharmacokinetic profiles, lack of human toxicity, and specific antimicrobial properties (*AshaRani, Hande & Valiyaveettil, 2009*). It is of paramount interest to the pharmaceutical manufacturers that the overall process of producing nanoparticle systems is ecologically balanced while being cost-optimized (*Gade et al., 2008*; *Ouda, 2014*). Contrary to the traditional synthetic methods, biological methods of generating nanoparticles are quite adaptive to the environment and also cost-effective (*Govindaraju et al., 2010*). The most important merit of biologically synthesised nanoparticles is their non-toxic nature and easy biological metabolism. These advantages have made biologically derived nanoparticles one of the most emerging formulation designs widely accepted in the pharmaceutical ecosystem (*Das & Smita, 2018*).

In recent years, plants have been widely explored for their active principles to treat complex ailments. Novel phytoconstituents derived from plant sources are spanning again around the pharmaceutical markets to develop new drugs, and among these, one of the most commonly used medicinal plants is Holy Basil, that is, *Ocimum sanctum* L. (*O. sanctum*). This plant has been used for traditional therapy and also reported as having medicinal significance for anticancer, antimicrobial, cardio-protective, antidiabetic, analgesic, antispasmodic, antiemetic, hepatoprotective, and antifertility actions. Leaves of the *O. sanctum* L. contain eugenol (1-hydroxy-2-methoxy-4-allylbenzene) as a major active chemical constituent and have been proven for their therapeutic efficacy in various ailments in modern clinical practise (*Hemaiswarya, Kruthiventi & Doble, 2008*; *Raseetha, Cheng & Chuah, 2009*). There are various studies on the synthesis of nanoparticles through the green route using parts of plant extracts such as tea leaves, the stem bark of *Callicarpa maingayi*, *Terminalia chebula*, *Papaver somniferum,* and *Aloe vera*. Additionally, silver nanoparticles
have been reported for anti-angiogenesis, anti-inflammatory, anti-platelet, anti-bacterial, and antiviral activity (*Bindhani & Panigrahi, 2015*).

The misuse of antimicrobials during the last two decades has increased the existence of antibiotic resistance in almost all bacterial strains including *A. baumannii*, *Klebsiella pneumoniae*, *Pseudomonas aeruginosa* and *Staphylococcus aureus*. This has not only made several anti-microbial drugs worthless, but it has also compelled the researchers to explore alternative solutions for fighting deadly microbial infections (*Nikaido, 2009*). Hence, some recent studies have focused on using silver nanoparticles (AgNPs) as one of the alternatives and have proven the antimicrobial property of silver nanoparticles against both Gram-negative and Gram-positive bacteria without any cytotoxic signs (*Biel et al., 2011*; *Lazar, 2011*). *Acinetobacter baumannii* is a Gram-negative, opportunistic bacterium that causes nearly 2–10% of all hospital-associated infections, particularly among immunocompromised patients (*Karlowsky et al., 2003*) The organism initially considered of low medical interest, *A. baumannii,* is now among the top hospital-associated pathogens isolated from the clinical settings, causing a wide array of infections like ventilator-associated pneumonia, septicemia, urinary tract infection, wound, skin, and soft tissue infections and often associated with high morbidity and mortality rates (*Karakoc et al., 2013*. The major challenges with *A. baumannii* are its extraordinary abilities to quickly develop resistance against new drugs, to form biofilm on abiotic surfaces, which helps them to survive on hospital equipment for long periods, and also to tolerate the harsh environment for survival (*Djeribi et al., 2012*). *A. baumannii* is considered a Red Alert human pathogen and is ranked the number one critical pathogen with high antibiotic resistance by the World Health Organisation for research and new drug discovery (*World Health Organization, 2017*).

This study describes an easy, fast, and simple method for the biosynthesis of AgNPs from *O. sanctum* leaf extract. We attempted to characterise the biosynthesized nanoparticles and evaluate their antibacterial activity against multidrug-resistant *A. baumannii* (MDR-*A. baumannii*).

## MATERIALS AND METHODS

This collaborative research study involved the preparation, characterization, and evaluation of the antimicrobial potential of biosynthesized AgNPs of *O. sanctum* leaf extract by adapting a green synthesis process. Figure 1 provides a schematic representation of the steps involved in this study.

### Materials

*O. sanctum* leaves were collected from the medicinal plant garden of the Delhi Pharmaceutical Sciences and Research University, New Delhi (verified by the CSIR-National Institute of Science Communication and Policy Research, New Delhi, authentication no. NIScPR/RHMD/consult/2022/4040-41). Silver nitrate was purchased from LobaChemie Private Limited, Mumbai, India. Antimicrobial susceptibility tests against multidrug-resistant *A. baumannii* were performed at the Sikkim Manipal Institute

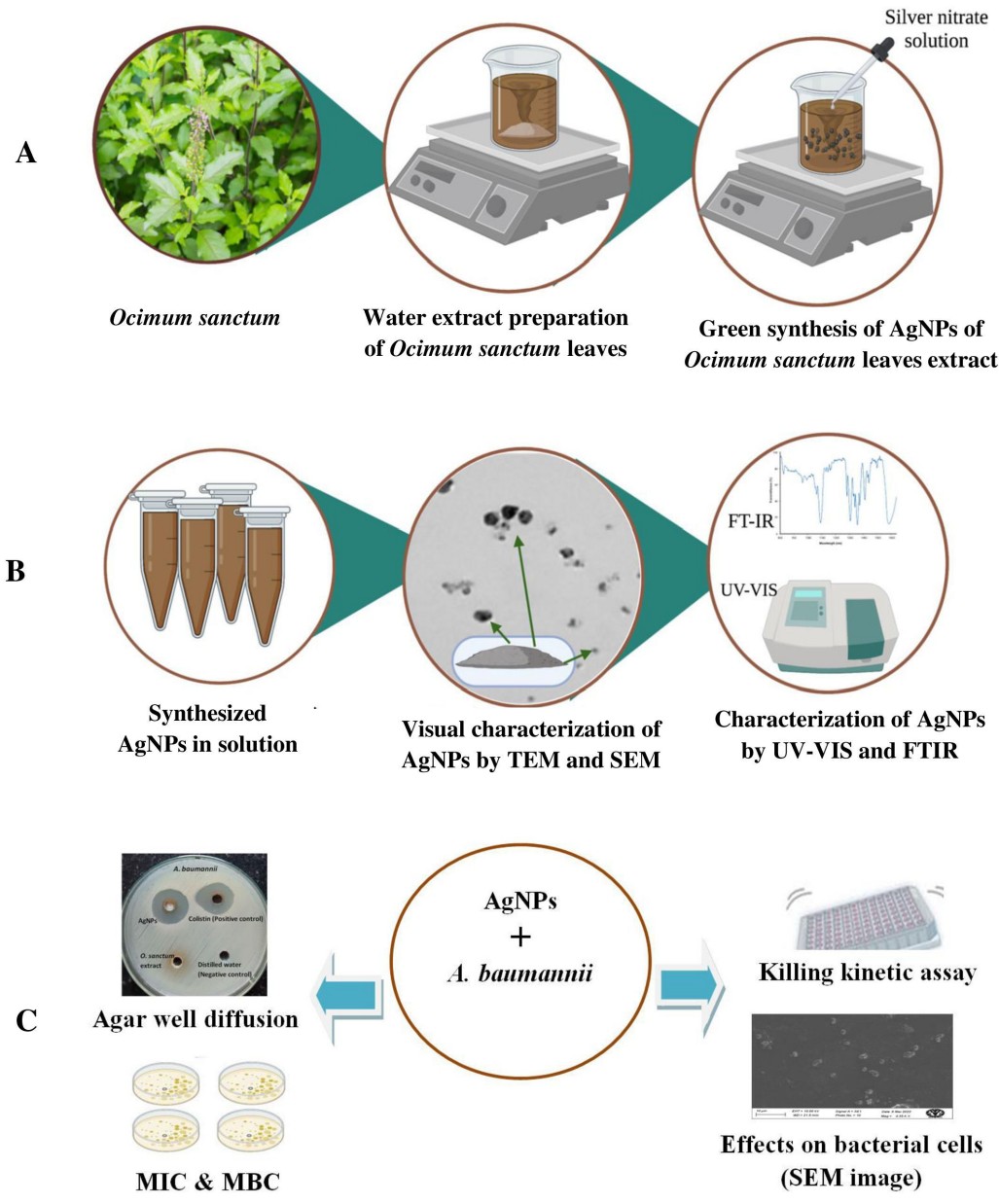

**Figure 1** Schematic representation of (A) synthesis, (B) characterization and (C) antimicrobial susceptibility testing of AgNPs of *O. sanctum.*

of Medical Sciences, Sikkim, India, under the ethical clearance number SMIMS/IEC/2019-29.

## Methods

### Plant collection and identification

*O. sanctum* is a relatively small, erect subshrub that reaches up to 60 cm in height and has reverse green or purple leaves and a hairy stem. The leaves are ovate, up to 5 cm long, toothed, and have a petiole (*Pattanayak et al., 2010*). For the present study, fresh leaves were

collected in September 2021 from the medicinal plant garden of the Delhi Pharmaceutical Science and Research University (DPSRU), New Delhi, and stored in airtight paper bags for further processing.

### Preparation of O sanctum leaves aqueous extract

To prepare the aqueous extract of leaves, fresh leaves were collected and placed in a beaker, washed with distilled water until completely free from dust or other residues, and then washed with ultrapure water (Milli-Q® HX 7000 SD, Merck, Sydney, Australia). A total of 25g of washed *O. sanctum* leaves were chopped into fine pieces and crushed in 100 mL pure water (Milli-Q®) using a mortar and pestle. The aqueous extract was ground and then boiled for 10 min at 80 °C in a 250 mL beaker. The aqueous leaf extract was then allowed to cool down to room temperature 25 °C and then filtered with Whatman filter paper with a 1.5 μm pore size (GE Healthcare Life Science, Karnataka, India) (*Singh et al., 2018*). These filtered leaf extracts were collected and stored at 4 °C, until further use (*Rao et al., 2013*).

### Qualitative phytochemical analysis of O. sanctum extract

The presence of amino acids and proteins was confirmed by boiling the *O. sanctum* leaf extract with a 0.2% Ninhydrin solution (0.2 g of ninhydrin dissolved in 100 mL ethanol) for five minutes. A change in colour towards violet indicates the presence of amino acids and proteins. The identification of phenols was done by mixing 1 mL of *O. sanctum* leaf extract with 2 mL of 99.5% ethanol and 3 drops of a 2% ferric chloride ($FeCl_3$) solution. The reddish-black colour confirmed the presence of phenols. The presence of flavonoids was tested by mixing 0.5 mL of *O. sanctum* leaf extract with a 2% sodium hydroxide (NaOH) solution and adding four drops of 1% hydrochloric acid (HCl). The colour changes from brown to yellow confirmed the presence of flavonoids (Table 1) (*Borah & Biswas, 2018*; *Saranya, Noorjahan & Siddiqui, 2019*).

### Preparation of 1 mM silver nitrate solution

The stock solution was prepared by weighing 170 mg of silver nitrate (99% extra pure) (LobaChemiePvt. Ltd., Mumbai, India) and dissolving it in 10 mL of ultra-pure water (Milli-Q® HX 7000 SD, Merck, Sydney, Australia). About 1 mL of this stock solution was taken and further dissolved in 100 mL of ultra-pure water. This solution was stored in an amber coloured bottle to prevent the self-oxidation of silver nitrate solution (*Saifuddin, Wong & Yasumira, 2009*).

### Green synthesis of silver nanoparticles (AgNPs)

Silver nanoparticles (AgNPs) were prepared by a single-step synthesis previously reported (*Ramteke et al., 2013*). In the process, a 90-mL solution of silver nitrate ($AgNO_3$) at 1 mM concentration was placed in a beaker on a magnetic stirrer (Remi, Mumbai, India) at 400 rpm, and 10 mL of an aqueous extract of *O. sanctum* leaves was added dropwise for half an hour (30 min). The colour of the solution turned from reddish brown to dark brown, indicating the formation of AgNPs. The preparation was kept at room temperature (25 °C) for 1 h. All procedures were performed in a dark room to prevent the oxidation of silver

**Table 1  Phytochemical analysis *O. sanctum* leaves extract.**

| Tests | Images | Results |
|---|---|---|
| Phenols | | Positive (black ring formation) |
| Flavonoids | | Positive (color changed to yellow) |
| Proteins and aminoacids | | Negative (No violet color change) |

nitrate. Nanoparticles were separated by the process of centrifugation at 3,000 rpm for 10 min (Remi, Mumbai, India). The synthesised nanoparticles were stored at 4 °C until further use.

### Characterization of the synthesized AgNPs of O. sanctum

The biosynthesized AgNPs were characterised by various parameters, as represented in Fig. 1. The process of silver ions reduction and biosynthesis of silver nanoparticles were continuously monitored every 20 min immediately after the blending of plant extract and silver nitrate solution, using an UV-Vis spectrophotometer (Shimadzu, Tokyo, Japan) at 452 nm. The absorption spectra of synthesised AgNPs of *O. sanctum* and the leaf extract of *O. sanctum* were also studied in the range of 200–800 nm. In addition, the average particle size and zeta potential were determined by dynamic light scattering, using the Litesizer™ 500 (Anton Paar, Buchs, Switzerland). The morphology of AgNPs in *O. sanctum* was visualised using a scanning electron microscopy (SEM, Leo 435 VP 501B; Philips, Texas, USA), with accelerated voltage up to 30 kV and magnification efficacy ranges from 10x to 300,000x. Following this, the synthesised nanoparticles were also characterised by

Transmission Electron Microscopy (TEM; JEOL, Tokyo, Japan) using a copper grid coated with carbon film and phosphotungstic acid (1% w/v) as a negative stain, and then air dried and allowed to rest at room temperature (25 °C). Next, samples were examined by Fourier-transform infrared spectroscopy (FTIR; PerkinElmer; Waltham, MA, USA). FTIR analysis provides information about the functional groups present.

## Stability testing of AgNPs of *O. sanctum*
### pH stability of AgNPs of O. sanctum:
pH stability of AgNPs of *O. sanctum* was evaluated by measuring the changes in particle size at different pH values (pH 2-9). A volume of 5 mL of freshly prepared AgNPs of *O. sanctum* was mixed with an equal volume of de-ionized water and then adjusted to the desired pH values (pH 2-9) using 1.0 M NaOH or 1% HCl. After that, the particle size of the silver nanoparticles was determined by the Malvern instrument (Zetasizer, Nanoseries) at 25 °C (*Wang et al., 2021*).

### Stability testing of AgNPs of O. sanctum at different concentration of sodium chloride (NaCl):
The effect of mineral ions ($Na^+$ and $Cl^-$) on the stability of AgNPs of *O. sanctum* was evaluated by measuring changes in particle size at different salt concentrations (NaCl: 0.05, 0.1, 0.2, 0.3, and 0.5 M). Freshly prepared, 5 mL AgNPs of *O. sanctum* were mixed with an equal volume of de-ionized water at pH 7. Different concentrations of sodium chloride solution were then added to the AgNPs dispersion. After 2 h, the particle size of AgNPs from *O. sanctum* was determined by the Malvern instrument (Zetasizer, Nanoseries) at 25 °C (*Wang et al., 2021*).

## The effects of biosynthesized AgNPs of *O. Sanctum* in bacterial cultures of *Acinetobacter baumannii*
### Bacterial broth preparation
A pure culture of multidrug-resistant (MDR) *Acinetobacter baumannii,* isolated from the respiratory tract specimen of a patient attending the Central Referral Hospital, Gangtok, was identified by the Vitek®-2 system (BioMerieux, Marcy-l'Étoile, France) and further confirmed by real-time PCR. The *A. baumannii* was further subcultured in a Muller-Hinton broth medium (Hi-Media, Mumbai, India) at 37 °C in the dark for 18–24 h. Before use, this bacterial broth was diluted in Muller-Hinton broth and adjusted to 0.5 McFarland turbidity ($10^8$ CFU/mL) using Densichek (BioMerieux, Marcy-l'Étoile, France). This bacterial broth was further tested for susceptibility to AgNPs of *O. sanctum* by different methods (Fig. 1).

### Agar well diffusion method
The antimicrobial susceptibility of the biosynthesized AgNPs was tested on *A. baumannii* in a Muller Hinton Agar (MHA) plate (Hi Media, Mumbai, India). First, the 0.5 McFarland turbid broth of *A. baumannii* was inoculated by the lawn culture method using a sterile cotton swab on a MHA plate and let it air dry. Four holes of six mm diameter were made in the plate with the help of a sterile borer. A volume of 100 μL of AgNPs of *O. sanctum* at a concentration of 200 μg/mL and the same volume of aqueous extract of *O. sanctum* at 1,000 μg/mL were added to two holes. The remaining two holes were used for the positive

and negative controls. Antibiotic colistin at a concentration of 16 μg/mL was used as a positive control, and sterile double-distilled water was used as a negative control. The plate was then incubated at 37 °C in the dark for 18–24 h. After incubation, the zone of inhibition was measured with the help of a ruler (*Alzahrani et al., 2020*).

## MIC and MBC determination

MIC (minimum inhibitory concentration) and MBC (minimum bactericidal concentration) were determined by the microdilution method (*Dash et al., 2012*). *A. baumannii* was cultured in a nutrient broth medium (Hi-Media, Mumbai, India) at 37 °C in the dark for 18–24 h. Before use, the bacterial broth was diluted in Nutrient Broth and adjusted to 0.5 McFarland turbidity ($10^8$ CFU/mL) using Densichek (BioMerieux, Marcy-l'Étoile, France). Then, 100 μL of this broth was added to the seven wells of the microtiter plate. The AgNPs of *O. sanctum* were diluted in deionized water by serial dilution, ranging from 2–256 μg/ mL and 100 μL of these different concentrations of AgNPs of *O. sanctum* were added to the wells loaded with the bacterial broth. The microtiter plate was then incubated at 37 °C in the dark for 24 h. The MIC value was noted by observing the turbidity on microtiter wells due to the bacterial growth. The MIC value corresponded to the minimum concentration of AgNPs in *O. sanctum* that inhibited the visible growth of bacteria in microtiter wells.

The MBC was obtained by sub-culturing the bacteria on a sterile MHA plate from the microtiter wells without turbidity and incubated at 37 °C in dark for 24 h. The minimum concentration of AgNPs which completely killed and reflected no growth of bacteria on the MHA plate was considered as the MBC value. Therefore, the MBC value obtained corresponded to the minimum concentration of AgNPs of *O. sanctum* that restricted 100% of the bacterial growth. Similarly, the MIC and the MBC of the positive control colistin were determined.

## The time-kill kinetics assay

The killing kinetics assay of *A. baumannii* against AgNPs of *O. sanctum* was performed spectrophotometrically (Shimadzu, Tokyo, Japan) at OD 600 nm in triplicates. *A. baumannii* was cultured in a nutrient broth medium (Hi-Media, Mumbai, India) at 37 °C in the dark for 18–24 h. Before use, the bacterial broth was diluted in Nutrient Broth and adjusted to 0.5 McFarland turbidity ($10^8$ CFU/mL) using Densichek (BioMerieux, Marcy-l'Étoile, France). A volume of 100 μL of this broth was added to the five wells of the microtiter plate. Then, 100 μL of AgNPs from *O. sanctum* at concentrations of 32 μg/mL (MIC), 64 μg/mL (MBC), 128 μg/mL, 256 μg/mL and 512 μg/mL were added to the five wells loaded with the bacterial broth. The microtiter plate was then incubated at 37 °C in the dark, and the bacterial viability was measured spectrophotometrically in triplicates at 0, 2, 4, 8, 12, 18, and 24 h of incubation. The negative control (bacterial broth without AgNPs and without antibiotic colistin) and positive control (bacterial broth treated with antibiotic colistin at a MIC of 2 μg/mL) were included in the test. The percentage of inhibition of bacterial growth was calculated in comparison with the negative control (*Das et al., 2017*), and a statistical correlation was made with the positive control.
### Effects of biosynthesized silver nanoparticles on *A. baumannii* cells

A volume of 10 mL of *A. baumannii* in nutrient broth medium with a concentration of $10^8$ CFU/mL was treated with the MIC value (32 μg/mL) of AgNPs of *O. sanctum* and incubated at 37 °C in the dark with shaking at 198 rpm for 12 h. A control experiment was performed in the absence of AgNPs. After incubating for 12 h, the bacterial culture tube was centrifuged at 3,000 rpm for 5 min, and the supernatant was discarded. The resulting pellets were fixed with 50 μL of 2.5% glutaraldehyde for 5 min at 37 °C and washed three times with 1X PBS. The pellets were then suspended in 50 μL 1X PBS and used to take images by scanning electron microscopy (SEM; Leo 435 VP 501B; Philip, Texas, USA) (*Das et al., 2017*).

### General cytotoxicity testing using the MTT assay

Human lung adenocarcinoma cell line A549 was obtained from the NCCS cell repository (Pune, India). A549 cells were cultured in DMEM supplemented with penicillin (100 g/mL), streptomycin (100 U/mL), and 10% heat-inactivated foetal bovine serum (FBS). Cells were maintained in a humidified atmosphere at 37 °C and 5% $CO_2$. For cytotoxicity studies with *O. sanctum* leaf extract and AgNPs of *O. sanctum*, A549 cells were seeded in appropriate cell culture dishes in DMEM/10% FBS one day prior to the exposure. Further, A549 cells were seeded in a 96-well tissue culture plate at a density of 5,000 cells per well. After 24 h, the cells were treated with different concentrations (0.97–500 μg/mL) of AgNPs of *O. sanctum* for 24 and 72 h. After treatment, the media was discarded, and MTT (3-(4,5-dimethylthiazol-2-yl)-2,5-diphenyltetrazolium bromide) at a final concentration of 0.5 mg/mL was added to each well. The plates were then incubated for two hours at 37 °C in a $CO_2$ incubator. After incubation, the media with MTT was discarded, and the formazan crystals formed were dissolved in DMSO at 37 °C for 15–20 min. The absorbance of dissolved formazan was measured at 570 nm, with a reference wavelength of 690 nm. Similarly, the test was also performed with *O. sanctum* extract. The control experiment was performed without AgNPs and *O. sanctum* extract. The resultant absorbance, which is directly proportional to cell viability, was converted into percent viability, and the viability of control cells was considered to be 100% (*Beer et al., 2012*; *Subha et al., 2021*).

### Statistical analysis

All data were recorded and analysed using the statistical package software IBM SPSS version 25 (SPSS, Chicago, IL, USA). The differences between mean values were tested for significance by one-way ANOVA analysis. A *p* value of <0.05 was considered statistically significant.

## RESULTS

### Extract preparation of *O. sanctum* leaves and qualitative phytochemical analysis

The aqueous extract of fresh leaves of *O. sanctum* collected from the medicinal plant garden of DPSRU, New Delhi, was prepared by boiling 25 g of leaves in 100 mL of Millipore water. Phytochemical tests confirmed the presence of phenols and flavonoids in the extract; however, proteins or amino acids were absent in the prepared extract (Table 1).

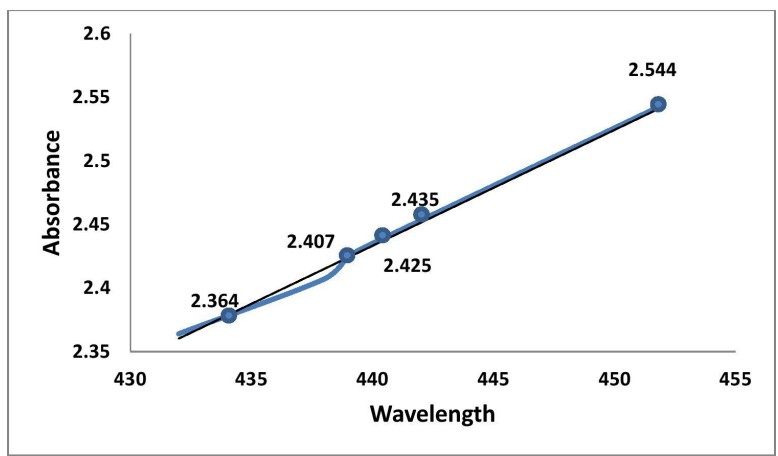

**Figure 2** Graphic representation of UV absorbance of AgNPs of *O. sanctum* at 452 nm (at 20 min time intervals).

## Preparation and characterization of silver nanoparticles of *O. sanctum*

Silver nanoparticles of *O. sanctum* were prepared using a single-step synthesis method, and further characterization was performed by spectrophotometry, zeta sizing, and electron microscopy.

### UV-visible (UV-Vis) spectroscopy

The silver surface plasmon resonance (SPR) was obtained by measuring the peak at 452 nm in the UV-Vis absorption spectrum of the reaction solution, which proved the formation of silver nanoparticles from silver ions. The broad band of UV-Vis absorption spectra is due to the presence of various metabolites of *O. sanctum* extract presented in the reaction solution that were read in the experimental spectrophotometric range (Figs. 2 and 3).

### Particle shape, size, and morphology

SEM images of prepared nanoparticles indicated that particles were almost spherical with smooth surfaces and a size range of 73.24–87.89 nm (Fig. 4). Morphological examination by TEM confirmed the spherical shape of most nanoparticles with a size range of 29–54.9 nm (Fig. 5), while some oval and/or elliptical-shaped nanoparticles were also formed, which is the common feature of most of the biologically synthesised nanoparticles. Lighter edges with a heavier centre were also visible, confirming the capping of protein biomolecules with AgNPs. The mean particle size of nanoparticles was 55 nm, which was fully concordant with results from TEM and SEM analysis. Particles showed a zeta potential of around −27 mV, and an increase in negative values confirmed the repulsion between particles, which also verified the stability of the formulation (Fig. 6A). Furthermore, the particle size distribution of AgNPs from *O. sanctum* showed 74.95 nm with a 27.9% polydispersity index (Fig. 6B).

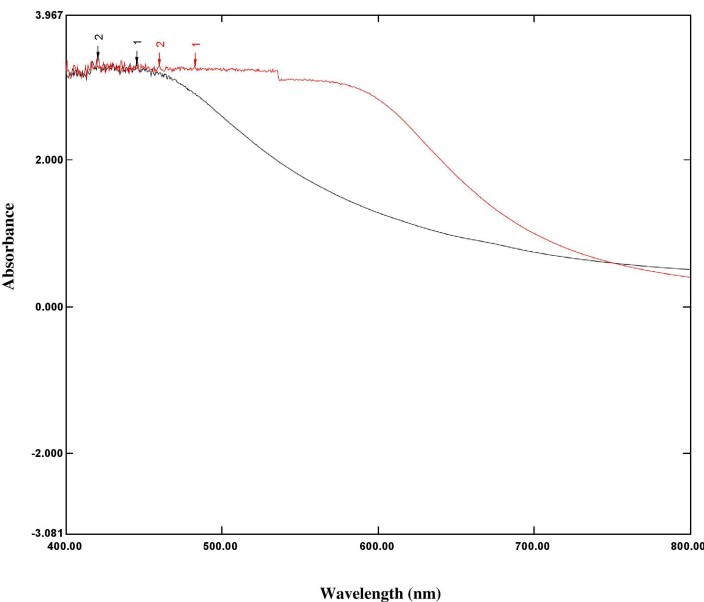

**Figure 3** Overlay graph represented the UV spectrum absorbance ranges 200 nm–800 nm of *O. sanctum* extract (black line) and AgNPs of *O. sanctum* (red line).

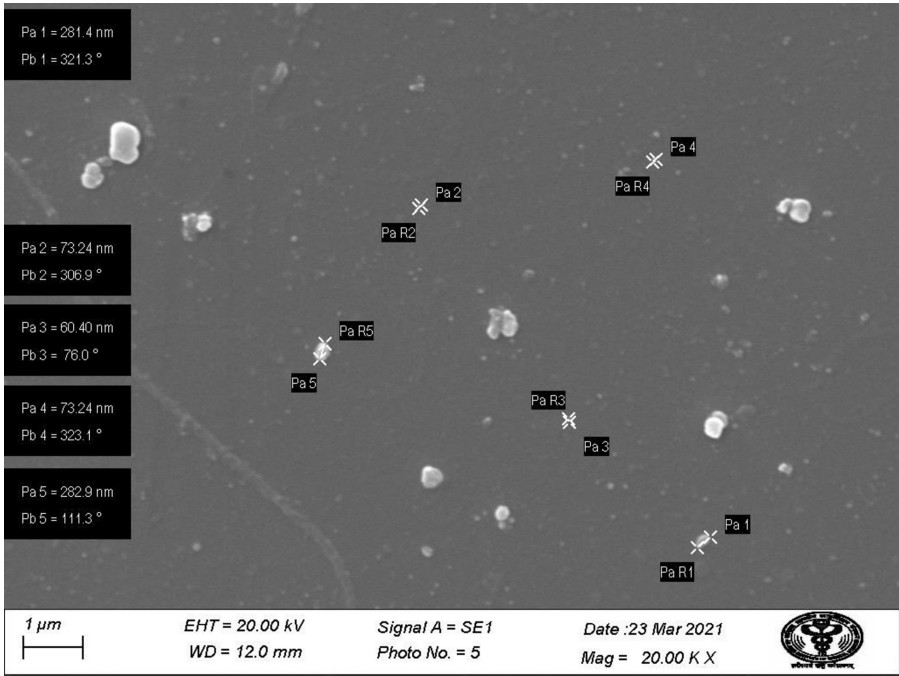

**Figure 4** SEM image showing particle range under 1 μm (20.00KX magnification with working distance (WD) 12.0 mm).

## Fourier transform infrared spectroscop (FTIR)

Silver nanoparticles (AgNPs) synthesised from *O. sanctum* are of interest due to their potential use in various biomedical applications. The FTIR spectra of aqueous leaf extract

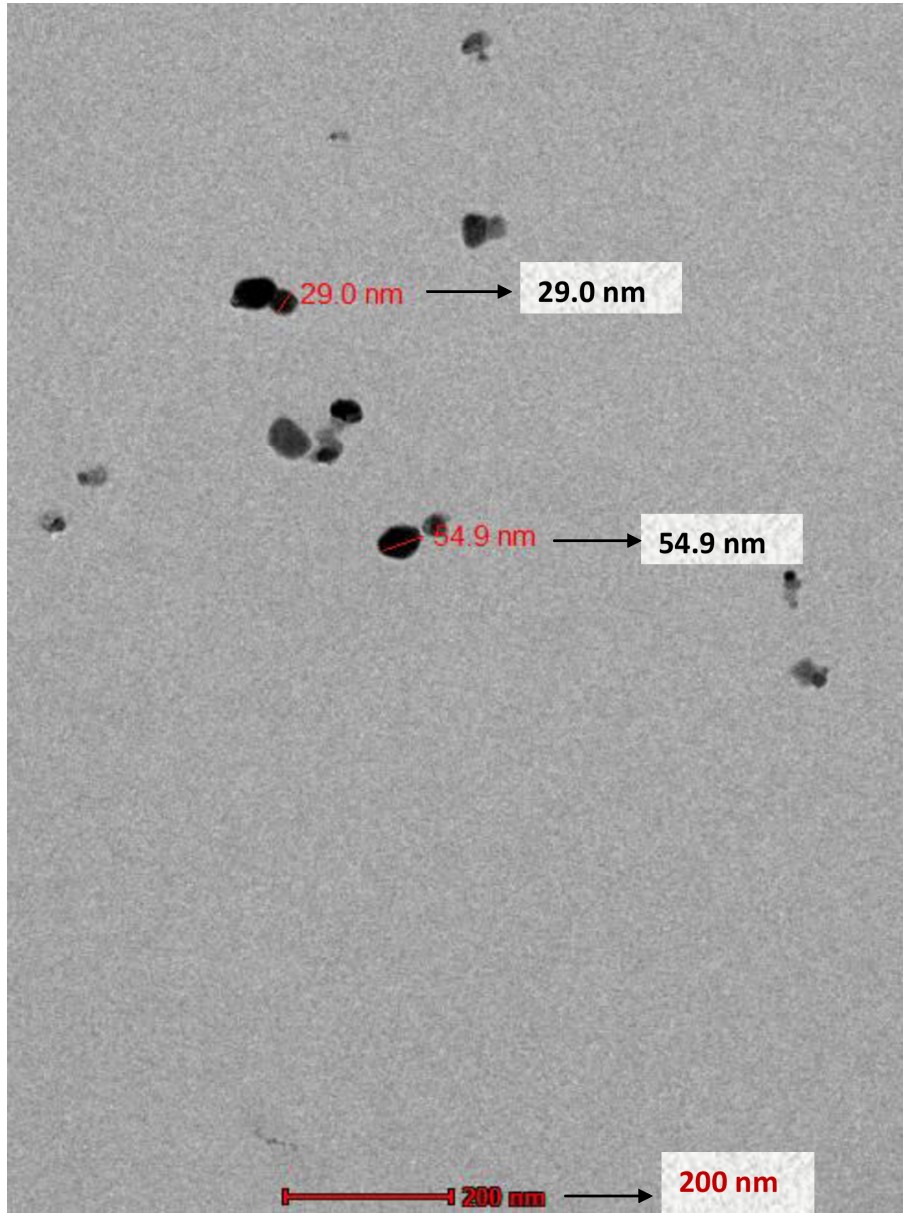

**Figure 5** TEM micrograph of AgNPs synthesized by the reaction of 0001 M silver nitrate with *O. sanctum* leaf extract.

and the biosynthesized AgNPs have been shown in Figs. 7A and 7B. The FTIR spectral analysis revealed strong peaks at 3,619.63 and 3,565.33, 1,741.28 and 1,693.51, and 650.74 and 651.24 cm$^{-1}$.

The strong bands at 3,619.63 and 3,565.33 cm$^{-1}$ indicate the presence of alcohols with free OH groups. The vibrational peaks at 1,741.28 and 1,693.51 cm$^{-1}$ represent the presence of an alkene group. The absorption bands at 650.74 and 651.24 cm$^{-1}$ are assigned

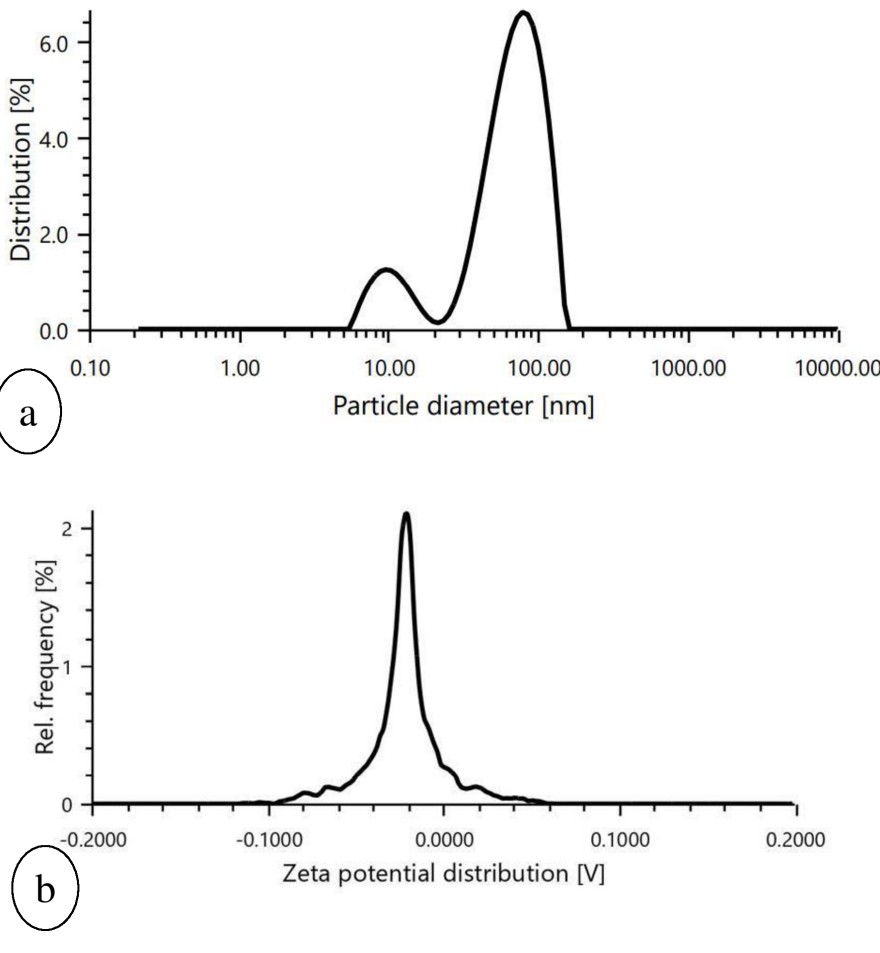

**Figure 6** Graphical representation of (A) particle size determination of biosynthesized AgNPs, (B) Zeta potential measurement of biosynthesized AgNPs.

to the aromatic C-H bending. The Fig. 7A represents the FTIR interpretation of *O. sanctum* leaves extract, and the Fig. 7B represents the FTIR interpretation of AgNPs of *O. sanctum*.

## Stability testing of AgNPs of *O. sanctum*

The pH stability testing for AgNPs of *O. sanctum* revealed no obvious changes in the particle size at pH 2 to 4, further increase in particle size was observed between the pH range from 5 to 9 (Table 2) (*Wang et al., 2021*). The influence of mineral ion concentrations was studied using sodium chloride solution at different concentrations (0.05 M to 0.5 M), and the analysis showed that in the range of ionic strength (0.05 to 0.2 M), AgNPs of *O. sanctum* were relatively stable in size. However, the nanoparticles became unstable when NaCl levels exceeded the salt concentration (0.3 to 0.5 M), causing the aggregation (Table 3).

## Antibacterial activity
### *Agar well diffusion method, MIC and MBC determination*
The zone of inhibition determined by the agar-well diffusion method after 24 h of incubation formed by AgNPs of *O. sanctum* (200 μg/mL) and colistin (16 μg/mL, positive

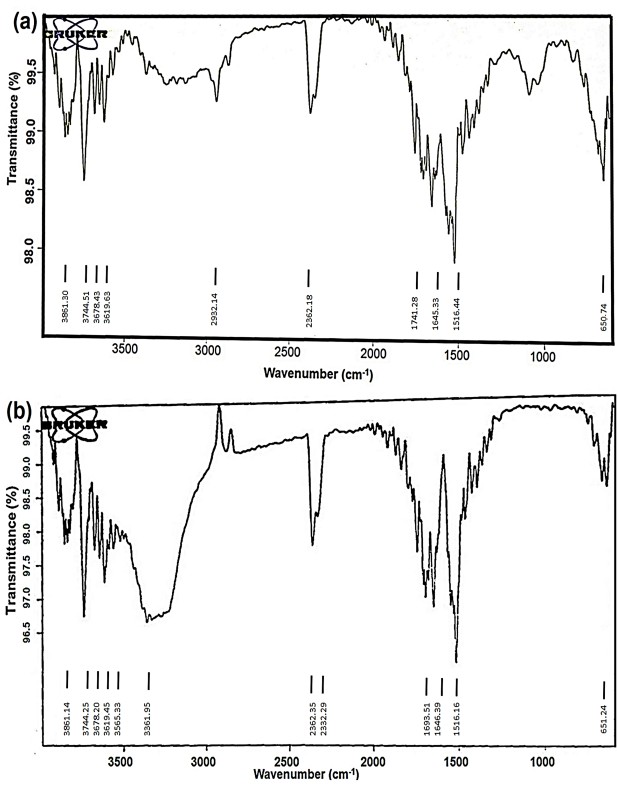

**Figure 7** FTIR spectra of (A) aqueous extract of *O. sanctum* leaves and (B) biosynthesized AgNPs of *O. sanctum*.

**Table 2 pH stability of AgNPs of *O. sanctum* leaves extract.**

| pH | Temperature | Particle size diameter (nm) | Polydispersity index |
| --- | --- | --- | --- |
| 2 | 25 °C | 0.000 | 0.263 |
| 3 | 25 °C | 0.000 | 1.000 |
| 4 | 25 °C | 0.000 | 1.000 |
| 5 | 25 °C | 59.91 | 1.000 |
| 6 | 25 °C | 65.69 | 0.765 |
| 7 | 25 °C | 55.51 | 0.661 |
| 8 | 25 °C | 84.35 | 0.338 |
| 9 | 25 °C | 138.6 | 0.662 |

control) was 15 mm and 14 mm, respectively, against the MDR *A. baumannii*. No zones were observed against distilled water (the negative control) or *O. sanctum* leaves extract (Fig. 8). The MIC and MBC of AgNPs of *O. sanctum* determined by the microdilution

**Table 3** Stability analysis of AgNPs of *O. sanctum* leaves extracts at different concentration of NaCl.

| Concentration of NaCl | Temperature | Particle size diameter (nm) | Polydispersity index |
|---|---|---|---|
| 0.05 M | 25 °C | 73.74 | 0.154 |
| 0.1 M | 25 °C | 71.98 | 0.164 |
| 0.2 M | 25 °C | 60.96 | 0.154 |
| 0.3 M | 25 °C | 33.15 | 1.000 |
| 0.5 M | 25 °C | 40.51 | 1.000 |

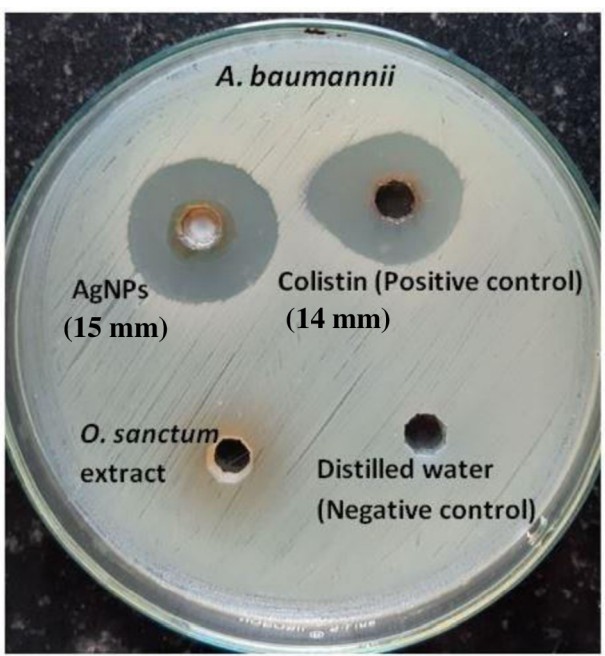

**Figure 8** Agar well diffusion test of AgNPs, colistin (positive control) and *Ocimum sanctum* leaves extract against *A. baumannii*.

method against the MDR *A. baumannii* were 32 µg/mL and 64 µg/mL respectively. The MIC and MBC of positive control colistin were 2 µg/mL and 4 µg/mL respectively.

## The time kill kinetics assay

The bactericidal activity was observed gradually up to 12 h of incubation with the 64 µg/ml MBC and higher concentrations of AgNPs, and the complete killing was observed within 24 h. The result showed time-dependent and gradual inhibitory and bactericidal activity against MDR *A. baumannii*. The AgNPs at their different concentrations (32 µg/mL, 64 µg/mL, 128 µg/mL, 256 µg/mL and 512 µg/mL) and at different time intervals showed significant killing actions against MDR *A. baumannii*. A statistical comparison was only done with the 32 µg/mL (MIC) of AgNPs and 2 µg/mL (MIC) of colistin. Comparison using 32 µg/mL of AgNPs (MIC) at 4, 12, and 24 h with 2 µg/mL of colistin (MIC) showed statistically significant results ($P < 0.05$). Though, the statistical comparison was done

**Table 4 Time kill kinetics assay results for viability of MDR *A. baumannii* at different concentrations of AgNPs of *O. sanctum* leaves extract and at different time interval.**

| AgNPs (µg/ml) | Duration of test (hour) | | | | | | | |
|---|---|---|---|---|---|---|---|---|
| | 0 h | | 4 h | | 12 h | | 24 h | |
| | Abs ± SD | PV (%) | Abs ± SD | PV (%) | Abs ± SD | PV (%) | Abs ± SD | PV (%) |
| 32 | 0.036 ± 0.0 | 100 | 0.038 ± 6.2 | 52.05[*] | 0.105 ± 1.16 | 49.06[*] | 0.221 ± 0.56 | 63.50[*] |
| 64 | 0.042 ± 7.7 | 104.24 | 0.034 ± 0.28 | 43.03 | 0.028 ± 0.68 | 12.72 | 0.019 ± 0.37 | 5.36 |
| 128 | 0.050 ± 4.0 | 98.81 | 0.029 ± 1.57 | 33.33 | 0.025 ± 0.34 | 10.96 | 0.018 ± 0.28 | 4.97 |
| 256 | 0.068 ± 3 | 99.04 | 0.050 ± 1.01 | 47.61 | 0.043 ± 0.60 | 17.47 | 0.035 ± 0.49 | 9.21 |
| 512 | 0.112 ± 7.1 | 100.19 | 0.095 ± 1.24 | 63.75 | 0.090 ± 0.75 | 31.03 | 0.078 ± 0.36 | 19.9 |
| Colistin (2 µg/mL) | 0.036 ± 1.6 | 100.95 | 0.034 ± 2.49 | 46.57 | 0.109 ± 1.56 | 50.93 | 0.210 ± 0.54 | 57.37 |

**Notes.**
[*]Indicate significant at *P* value <0.05 in comparison with positive control colistin.
PV, Percent viability; Abs, Mean Absorbance; SD, Standard Deviation.
Statistical comparison was only done between MIC value of AgNPs (32 µg/mL) and MIC value of positive control colistin (2 µg/mL).

with two different concentrations, but both of them represent the MIC against same *A. baumannii* isolate. Table 4 represent the details of killing action at different concentrations and time interval.

### Effects of biosynthesized silver nanoparticles on *A. baumannii* cells

Electron micrographs by SEM of *A. baumannii* cells, untreated and treated with AgNPs, are shown in Fig. 9. SEM images of untreated *A. baumannii* showed a typical clear surface structure with smooth and intact cell morphology, whereas those of *A. baumannii* treated with AgNPs showed severely damaged cell structure with ruptures, gaps, an irregular surface, and fragments.

### Cytotoxicity test using MTT assay

Cytotoxicity against the human lung adenocarcinoma cell line A549 at concentrations of 500 µg/mL and 250 µg/mL of both AgNPs and *O. sanctum* extract showed that cells did not remain viable after 24 and 72 h of treatment. However, at all the other concentrations, ranging from 0.97 to 125 µg/mL A549 cells showed viability almost equivalent to that of untreated cells at both time points (Fig. 10).

### DISCUSSION

This study was an attempt to synthesise and characterise AgNPs of *O. sanctum* and further evaluate their potential as an antimicrobial agent against *A. baumannii*. Preliminary phytochemical screening of the *O. sanctum* extracts confirmed phenols and flavonoids in the leaf extract, while it was found negative for the presence of proteins. It is well established by previous studies that colour change occurs after addition of *O. sanctum* extract to silver nitrate solution during the reaction (*Pirtarighat, Ghannadnia & Baghshahi, 2019*), which indicates the reduction of silver ions to form nanoparticles with the extract metabolites. Further, the colour intensity also increases with respect to the standing time (*Fayaz et al., 2010*; *Kumar, Selvi & Govindaraju, 2013*). It is interesting that the ultraviolet and visible (UV-Vis) absorption spectra of the prepared mixture confirmed the silver nanoparticles

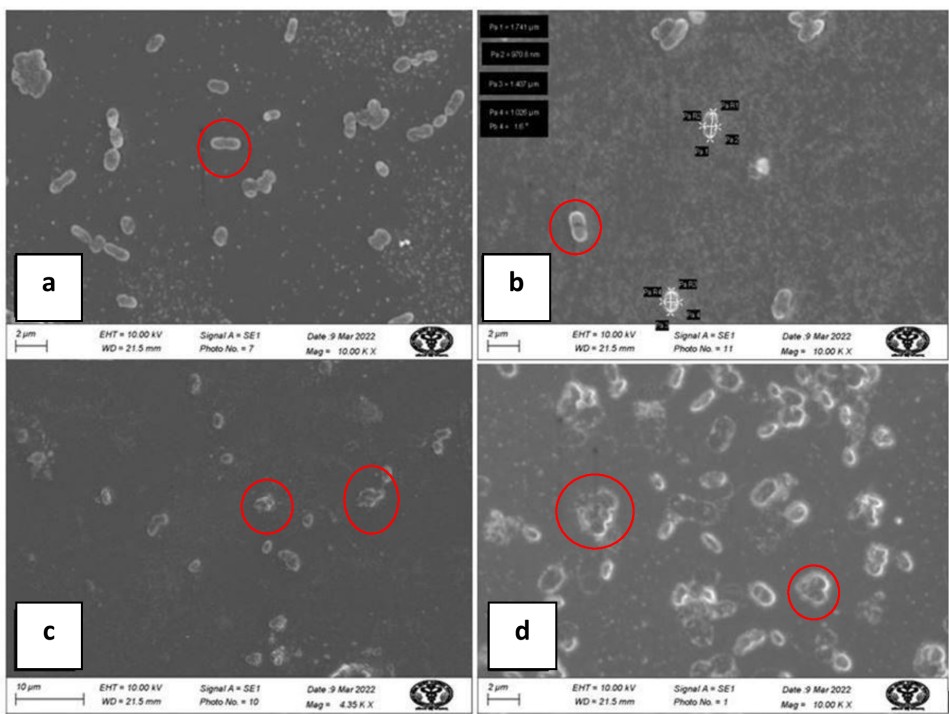

**Figure 9** SEM micrograph of *A baumannii* before and after treatment with AgNPs; (A, B) clear, smooth cells before treatment (inside red circle); (C, D) damaged, ruptured cells after treatment (inside red circle).

formation from silver ions, with a peak at 452 nm. This finding, supported by the broad band of UV absorption, is mainly due to the presence of organic metabolites in the *O. sanctum*- based aqueous extract (*Rao et al., 2013*). Furthermore, SEM and TEM analysis of biosynthesized nanoparticles (Figs. 4 and 5) demonstrated that nanoparticles are almost spherical in shape with smooth surfaces, have a mean particle size of 55 nm with a −27 mV zeta potential showing stable particles, and their particle size distribution is 74.95 nm. A previous study showed that the average size of silver nanoparticles biosynthesized using the leaf extract of *O. sanctum* was 42 nm (*Rao et al., 2013*).

The FTIR spectra depicted some extent of shifting of the AgNPs spectra over that of the aqueous extract, which might be due to the presence of functional groups present in the biosynthesis of plant extract and the capping of nanoparticles. It also exhibited biosynthesized AgNPs and an aqueous extract that differed very slightly in their absorption bands (Figs. 7A and 7B). This may be illustrated by the fact that available biomolecules in plants play a crucial role in the reduction of metal ions and the formation of small nanoparticles (*Kandasamy et al., 2013*). In addition, peaks at 3,339.30 cm$^{-1}$ and 1,634.70 cm$^{-1}$ indicated the binding of proteins, carbohydrates, and nitrogenous compounds on the surface of nanoparticles (*Das & Smita, 2018*).

The stability of AgNPs of *O. sanctum* tested at different pH levels showed changes in particle size after pH 5 to 9, and results at different concentrations of NaCl showed that

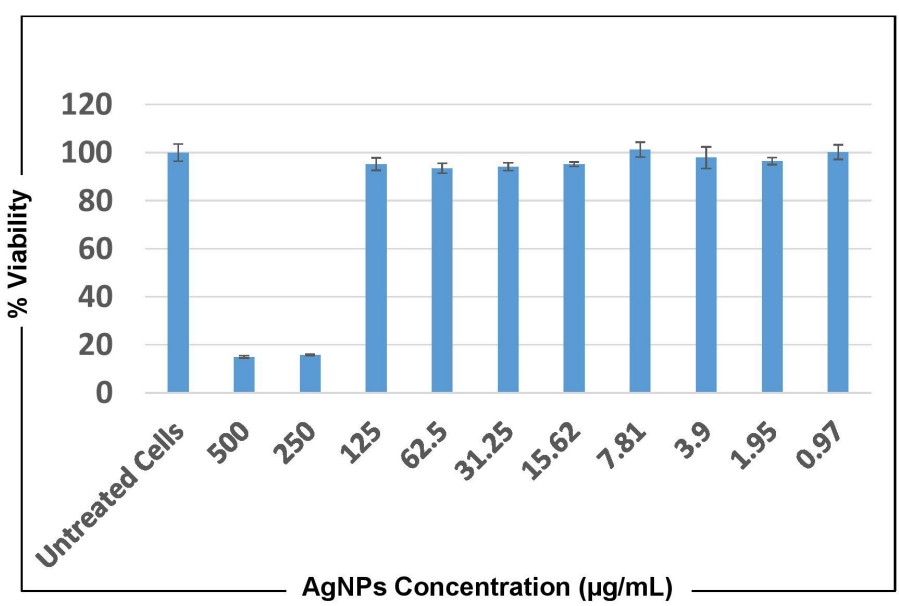

**Figure 10** MTT cytotoxicity test results representing cell viability upon treatment with AgNPs of *O. sanctum* for 24 h.

the particle size was reducing, which was in par with a previous study (*Wang et al., 2021*). It is imperative that the *O. sanctum* extract play a dual role as a reducing agent as well as a stabilising agent for AgNPs This is therefore highly recommended for more comprehensive studies to justify this association and come up with the final conclusion.

The antimicrobial studies determined that there was no zone of inhibition found in tulsi extract or sterile distilled water, but AgNPs from *O. sanctum* have promising and effective antibacterial activity on *A. baumannii*, showing a 15 mm zone of inhibition. The MIC and MBC results also proved that the AgNPs were able to inhibit the growth of MDR *A. baumannii*. The antibacterial activity can be explained based on nanoparticles interaction with microorganisms (*Franci et al., 2015*) by the released silver ions being attached to the cell wall of bacteria, modulating the cell membrane permeability, respiration blockage (*Dhas et al., 2014*; *Manjumeena et al., 2014*; *Muhsin & Hachim, 2014*), and destabilisation of the bacterial outer membrane and plasma membrane degradation followed by reduction of intracellular ATP (*Ajitha, Reddy & Reddy, 2014*; *Pirtarighat, Ghannadnia & Baghshahi, 2019*). Silver ions also have a great affinity to interact with the sulphur or phosphorus in cell biomolecules, ultimately ceasing bacterial replication (*Umashankari et al., 2012*). AgNPs might also have affected some of the cellular components and induced the damage of the cell membrane, which finally results in cell decomposition and death (*Li, Xie & Shi, 2010*).

The bactericidal activities of biosynthesized AgNPs from the extracts of other *Ocimum* species were also reported. *Tailor et al. (2020)* observed the antibacterial activity of AgNPs prepared from *Ocimum canum* against *Escherichia coli*, with a minimum zone of inhibition of 17 mm at 10 ppm concentration of AgNPs, while the maximum zone of inhibition of 24.5 mm was observed at 30 ppm concentration (*Tailor et al., 2020*). The susceptibilities

of 15 mm, 13 mm, and 12 mm were observed against *Bacillus vallisomortis, Bacillus subtilis,* and *Escherichia coli,* respectively, using AgNPs synthesized from *Ocimum bacilicum* (*Pirtarighat, Ghannadnia & Baghshahi, 2019*). Using the biosynthesized AgNPs from the extract of *O. gratissimum*, *Das et al. (2017)* noted no zone of inhibition in silver nitrate solution alone, but the bio-reduced AgNPs showed considerable growth inhibition against pathogenic *Escherichia coli* and *Staphylococcus aureus*. They observed a zone size of eight mm and 12 mm against *Escherichia coli* using 4 μg/mL (MIC) and 8 μg/mL (MBC) of AgNPs, respectively. Similarly, the zone sizes of 10 mm and 16 mm were observed against *Staphylococcus aureus* using 8 g/mL (MIC) and 32 μg/mL (MBC). The combined activity of phytochemicals from *Ocimum gratissimum* and AgNPs demonstrated a beneficial role in reducing the dose required for total microbial growth inhibition.

The killing kinetic assay showed time- and dose-dependent action against MDR *A. baumannii.* The bactericidal activity was gradual, and complete killing was observed within 24 h using MBC and higher concentrations of AgNPs. A statistically significant result was observed in comparison to the killing action of the antibiotic colistin. The time kill curve analysis by *Das et al. (2017)* using AgNPs against *Escherichia coli* and *Staphylococcus aureus* showed bacterial killing activity that increases with time of exposure of bacteria to AgNPs at their respective MBC concentrations, and complete bactericidal results were obtained. The bacterial exposure with AgNPs demonstrated a rapid dose and time-dependent killing leading to an early stationary phase. This rapid bactericidal activity of AgNPs could significantly decrease the bacterial mechanism to induce resistance development. Therefore, AgNPs may be a promising alternative to significantly reduce the development of drug resistance in bacteria and an effective antimicrobial agent for human use after the strong clinical trials (*Thammawithan et al., 2021*).

Various studies have reported that the bactericidal activity of AgNPs depends on their size and shape. The smaller the size, the higher would be the antibacterial property compared to the big size particles (*Panacek et al., 2006*). This result could be due to the higher penetration ability of smaller nanoparticles (*Morones et al., 2005*). This evidence of small size nanoparticles showed good results against bacterial inhibition, but studies also reported the adverse effects and health issues of nanoparticles due to their nano size. (*Basavaraja et al., 2008*; *El-Ansary & Al-Daihan, 2009*). This small size of nanoparticles makes them mobile both in the human body and in the external environment as well (*Braydich-Stolle et al., 2005*). *Pal, Tak & Song (2007)* demonstrated that truncated triangular AgNPs revealed the highest bactericidal activity against *Escherichia coli* when compared with rod- and spherical-shaped nanoparticles. The same result was also shown by *Sharma, Yngard & Lin (2009)*.

Apart from the antibacterial efficacy against MDR *A. baumannii*, cytotoxicity testing to the mammalian cell is also crucial to the development of novel antimicrobials. The optimum features that support the efficacy of a new antimicrobial agent require properties like potent antimicrobial activity and a low cytotoxicity level (*Thammawithan et al., 2021*). Our finding with the bio-synthesized AgNPs of *O. sanctum* showing antibacterial activity against MDR *A. baumannii* at a MBC of 64 μg/mL and cytotoxicity against the human A549 cell only above the concentration of 250 μg/mL clearly indicates that AgNPs are less to

moderately toxic against human cells compared to their effect on bacterial cells. This study shows a good ray of hope for the development of novel antibiotics using nanoparticles with more intense research and clinical trials.

## CONCLUSION

This study focused on the single-step green approach for the biosynthesis of silver nanoparticles from an aqueous leaf extract of *O. sanctum*. One of the most important benefits of this method is that it is eco-friendly and reduces traces of organic solvents that are hazardous to human health. Silver nanoparticles were successfully synthesized and confirmed by the colour change. The various evaluation parameters supported the nano-sized range with stable silver nanoparticles owing to the presence of biomolecules present in leaf extract that may act as surface-active stabilising agents supporting the formulation of silver nanoparticles. The antibacterial studies revealed their efficacy against clinically isolated MDR *A. baumannii,* and the cytotoxic activity of AgNPs and *O. sanctum* extract against mammalian cells showed moderate action. This effective bactericidal activity of AgNPs could lead to useful alternative treatment strategies to minimise bacterial resistance. Therefore, AgNPs may be a promising way to decrease the number of antibiotic-resistant bacteria and an alternative antimicrobial agent for human use after strong clinical trials. Further studies would be performed to prove its efficacy more effectively.

This study is a part of a PhD thesis entitled "Molecular characterization, detection of carbapenem resistance genes, and effect of natural products using nanotechnology against multidrug-resistant *Acinetobacter baumannii* isolated from various clinical specimens from Central Referral Hospital, Sikkim, India", Walailak University, Thailand.

### Funding

This work has been supported by the following: 1. Ph.D. Scholarship for Outstanding International Students, Scholarship no. MOE571900/110/2562 and Graduate Studies Research Fund CGS-RF-2021/02. Walailak University, Nakhon Si Thammarat, Thailand. 2. TMA Pai University Research Seed Grant-Major (2018-19), Reference no. 176/SMU/Reg/TMAPURF/30/2019. Sikkim Manipal University, Sikkim, India. 3. Project CICECO-Aveiro Institute of Materials, UIDB/50011/2020, UIDP/50011/2020 & LA/P/0006/2020, financed by national funds through the FCT/MCTES(PIDDAC). The funders had no role in study design, data collection and analysis, decision to publish, or preparation of the manuscript.

### Grant Disclosures

The following grant information was disclosed by the authors:
PhD. Scholarship for Outstanding International Students: MOE571900/110/2562.
Graduate Studies Research Fund: CGS-RF-2021/02.
Walailak University, Nakhon Si Thammarat, Thailand.

TMA Pai University Research Seed Grant-Major (2018-19): 176/SMU/Reg/TMA-PURF/30/2019.
Sikkim Manipal University, Sikkim, India.
CICECO-Aveiro Institute of Materials: UIDB/50011/2020, UIDP/50011/2020, LA/P/0006/2020.

## Competing Interests

Sonia M.R. Oliveira is an Academic Editor for PeerJ.

## Author Contributions

- Deepan Gautam conceived and designed the experiments, performed the experiments, analyzed the data, prepared figures and/or tables, authored or reviewed drafts of the article, and approved the final draft.
- Karma Gurmey Dolma conceived and designed the experiments, performed the experiments, analyzed the data, prepared figures and/or tables, authored or reviewed drafts of the article, and approved the final draft.
- Bidita Khandelwal conceived and designed the experiments, authored or reviewed drafts of the article, and approved the final draft.
- Madhu Gupta conceived and designed the experiments, performed the experiments, analyzed the data, prepared figures and/or tables, authored or reviewed drafts of the article, and approved the final draft.
- Meghna Singh conceived and designed the experiments, performed the experiments, analyzed the data, prepared figures and/or tables, authored or reviewed drafts of the article, and approved the final draft.
- Tooba Mahboob performed the experiments, analyzed the data, prepared figures and/or tables, authored or reviewed drafts of the article, and approved the final draft.
- Anil Teotia performed the experiments, analyzed the data, prepared figures and/or tables, and approved the final draft.
- Prasad Thota performed the experiments, analyzed the data, prepared figures and/or tables, and approved the final draft.
- Jaydeep Bhattacharya conceived and designed the experiments, analyzed the data, prepared figures and/or tables, and approved the final draft.
- Ramesh Goyal conceived and designed the experiments, analyzed the data, prepared figures and/or tables, and approved the final draft.
- Sonia M.R. Oliveira conceived and designed the experiments, authored or reviewed drafts of the article, and approved the final draft.
- Maria de Lourdes Pereira conceived and designed the experiments, authored or reviewed drafts of the article, and approved the final draft.
- Christophe Wiart performed the experiments, analyzed the data, authored or reviewed drafts of the article, and approved the final draft.
- Polrat Wilairatana performed the experiments, authored or reviewed drafts of the article, and approved the final draft.
- Komgrit Eawsakul conceived and designed the experiments, performed the experiments, authored or reviewed drafts of the article, and approved the final draft.

- Mohammed Rahmatullah performed the experiments, authored or reviewed drafts of the article, and approved the final draft.
- Shanmuga Sundar Saravanabhavan conceived and designed the experiments, analyzed the data, prepared figures and/or tables, authored or reviewed drafts of the article, and approved the final draft.
- Veeranoot Nissapatorn conceived and designed the experiments, performed the experiments, analyzed the data, prepared figures and/or tables, authored or reviewed drafts of the article, and approved the final draft.

## Ethics

The following information was supplied relating to ethical approvals (i.e., approving body and any reference numbers):

Institution Research Committee, Sikkim Manipal Institute of Medical Sciences, Sikkim Manipal University approved this study. (Ethical clearance no. SMIMS/IEC/2019-29)

## Data Availability

The raw data are available in the Supplementary Files.

## Supplemental Information

Supplemental information for this article can be found online at http://dx.doi.org/10.7717/peerj.15590#supplemental-information.

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
