# Peer review of "Green synthesis of silver nanoparticles using Ocimum sanctum Linn. and its antibacterial activity against multidrug resistant Acinetobacter baumannii"

_PeerJ, doi:10.7717/peerj.15590_

## Round 0.1 · original submission · Major Revisions

Three reviewers have now assessed your manuscript and found that there is room for improvement. For instance, your work would benefit from professional or fluent English-speaker revision. Also, conclusions are considered to be not fully supported by your findings, and the methodology is not properly defined in some instances. Therefore, I think it's worth responding to these concerns before further considering your work.

·

Basic reporting

Some issues were raised while reading through the article. A few I've noted below-
1. Introduction section should be improved.
2. Authors should be used more up-to-date references.
3. The paper contains some grammatical mistakes and syntax errors.
4. Conclusion section should be more specific.
5. If possible try to include graphical abstract.
6. Line 97: please list out a few bacterial-resistant strains

Experimental design

no comments

Validity of the findings

The conclusion portion has to be improvised.
Future prospects have to be mentioned.

Additional comments

References have to be updated, a few references are ancient.
In some of the references, the DOI is missing.
The grammatical mistakes have to be rectified.
few corrections are included in the file attached (go through proper punctuations)

·

Basic reporting

I was generally able to understand the writing, however, as English is not my native language, I was unable to give an opinion on whether it is written in a professional English language
I consider that there are figures that are not relevant, such as figure 1, 2 and 3, which could be condensed into one. The resolution of the figures should be improved, since the electron microscopy ones do not show the details described in the results section, likewise it is necessary to describe the figures in the figure captions more extensively. (see review file for particular details).

Experimental design

Gautam et al., present results of the characterization and antimicrobial effect of silver nanoparticles (AgNPs) obtained by green synthesis, I consider relevant for the publication, however, in my opinion some controls are needed, as well as there are elements that could lead to question the results, such as the calculation of the 1mM AgNO3 solution. In the attached document, I indicate throughout the writing the elements that should be corrected in order to strengthen the methodology.

Validity of the findings

I consider that it is necessary to include several controls in particular, the use of reference strains with well-defined susceptibility patterns that validate the antimicrobial susceptibility experiments
The statistical analysis presented must be justified based on the type of data obtained, especially it must be proven that they follow a normal distribution. The conclusions derived from the study are biased since the comparisons are not equivalent, that is, the antimicrobial effect of a substance in a lower concentration cannot be compared to the effect of another substance that is evaluated in a higher concentration. An alternative is proposed in the attached file.

Additional comments

no comment

·

Basic reporting

The authors describe the green synthesis of silver nanoparticles using an extract of ocimum sanctum. They characterize the nanoparticle and evaluate its effect.
The introduction could be improved by introducing the silver NPs in the paragraph sentence of line 70.
Just as it would be convenient to specify in the paragraph of line 83, which chemical compounds of the Ocimum sanctum are associated with therapeutic effects, since these would be expected to be present in the NPs.
I suggest you indicate the hypothesis and not only the objective, in order to establish the importance of each of your results.

Experimental design

It is suggested that the research question be defined to identify the relevance of the synthesis of silver NPs with the proposed plant, since in the literature there is a synthesis with other plant and fruit extracts.
It is recommended that within the methodology you mention the technique and the physicochemical interpretation that will be given.
Example: determination of particle size distribution by DLS; Stability of the nanostructure by Potential Z; Evidence of the formation of nanostructures or quantification through UV, etc.
Likewise, when describing the synthesis technique to guarantee its replication, it is suggested that they report defined concentrations for the reagents. In the case of the extract, they could quantify one of the compounds present to cover the concentration factor, which is decisive in a chemical reaction.
It would be recommended that in the methodology, the synthesis and characterization figures be integrated (Fig. 1 and 2)
And Figure 3 integrates all the processes described in the methodology: Diffusion, MBC and MIC, killing kinetics, cytotoxicity, SEM visualization of damage in bacteria.
Define your positive and negative controls, both for the characterization of the nanostructure and for the evaluation of its function.
In the methods it is recommended that the result is not described.
They could improve their micrographs to better identify their interpretation.

Validity of the findings

The validity of their findings cannot be tested as they are presented, as lack of controls were identified.
And their conclusions assume unverifiable statements, since for example they state that it is a profitable method, without presenting evidence.
Just as they mention that it is an innovative green synthesis method, when the synthesis reaction does not occur and there are other articles that follow the same synthesis procedure with extracts.
Therefore, it is suggested that they restate their conclusions.

Additional comments

These suggestions could help improve the understanding of the report.
Figure 4 would improve if the spectrum were presented before and after the formation of NPs. And this evidence could help the quantification in obtaining NPs.
Figure 5 and 6 could be improved to define the shape, size and morphology they are intended to demonstrate.
Figure 7 suggests that only the particle size distribution graph be presented and not the software report, and verify that the average offered by the equipment does not fall outside the distribution populations. Así mismo se recomienda que se de una adecuada interpretación de potencial Z, como estabilidad de estructura química o como estabilidad de la NP en la solución.
Figura 8. Se recomienda tener los controles adecuados para identificar los cambios de la formación de enlaces entre la Ag y los compuestos orgánicos del extracto.
In Figure 9, the controls of both the silver salt solution and that of an antibiotic-bactericide must be included.
In a kinetic it is recommended to start with an equal concentration of the substrate if the catalyst is going to be varied, or to keep the catalyst concentration constant and vary the substrate. Likewise, the kinetics over time is better visualized through a graph and not a table.
For a better visualization of the effect of the NPs on the bacteria, it is suggested to present a micrograph with treatment and another without treatment, unless they show a different approach, this suggestion is for figure 10.
For graph 11, a control with another silver NP would be missing to see if the effect is limited by silver or by the compound that is particularizing silver nitrate.

---

## Round 0.2 · accepted · Accept

I appreciate your having taken into account all concerns raised by reviewers.

·

Basic reporting

Corrections done

Experimental design

Corrections done

Validity of the findings

Corrections done

Additional comments

The author has resolved the comments in the revision.

·

Basic reporting

no comment

Experimental design

no comment

Validity of the findings

no comment

Additional comments

no comment